# Asymmetric Valleys: Beyond Sharp and Flat Local Minima

**Haowei He**[1][*]
hhw19@mails.tsinghua.edu.cn

**Gao Huang**[2,3]
gaohuang@tsinghua.edu.cn

**Yang Yuan**[1]
yuanyang@tsinghua.edu.cn
[1]Institute for Interdisciplinary Information Sciences, Tsinghua University
[2]Department of Automation, Tsinghua University
[3]Beijing National Research Center for Information Science and Technology (BNRist)

## Abstract

Despite the non-convex nature of their loss functions, deep neural networks are known to generalize well when optimized with stochastic gradient descent (SGD). Recent work conjectures that SGD with proper configuration is able to find wide and flat local minima, which are correlated with good generalization performance. In this paper, we observe that local minima of modern deep networks are more than being flat or sharp. Instead, at a local minimum there exist many asymmetric directions such that the loss increases abruptly along one side, and slowly along the opposite side – we formally define such minima as *asymmetric valleys*. Under mild assumptions, we first prove that for asymmetric valleys, a solution biased towards the flat side generalizes better than the exact empirical minimizer. Then, we show that performing weight averaging along the SGD trajectory implicitly induces such biased solutions. This provides theoretical explanations for a series of intriguing phenomena observed in recent work [25, 5, 51]. Finally, extensive empirical experiments on both modern deep networks and simple 2 layer networks are conducted to validate our assumptions and analyze the intriguing properties of asymmetric valleys.

## 1 Introduction

The loss landscape of neural networks has attracted great research interests in the deep learning community [9, 10, 32, 12, 15, 43, 36]. A deeper understanding of the loss landscape is important for designing better optimization algorithms, and helps to answer the question of when and how a deep network can achieve good generalization performance. One hypothesis that draws attention recently is that the local minima of neural networks can be characterized by their flatness, and it is conjectured that sharp minima tend to generalize worse than the flat ones [32]. A plausible explanation is that a flat minimizer of the training loss can achieve lower generalization error if the test loss is shifted from the training loss due to random perturbations. Figure 1(a) gives an illustration for this argument.

Although being supported by plenty of empirical observations [32, 25, 34], the definition of flatness was recently challenged in [11], which shows that one can construct arbitrarily sharp minima through weight re-parameterization without affecting the generalization performance. Moreover, recent evidences suggest that the minima of modern deep networks are connected with simple paths with low generalization error [12, 13]. It is empirically found that the minima found by large batch training

---

[*]Code available at https://github.com/962086838/code-for-Asymmetric-Valley

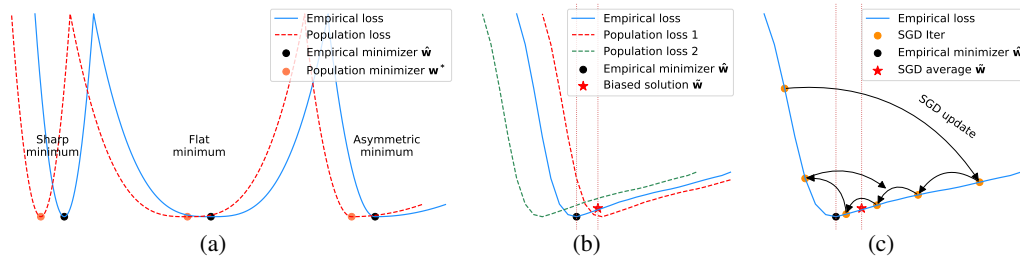

Figure 1: **(a)** An illustration of sharp, flat and asymmetric minima. If there exists a shift from empirical loss to population loss, flat minimum is more robust than sharp minimum. **(b)** For asymmetric valleys, if there exists a random shift, the solution $\tilde{w}$ biased towards the flat side is more robust than the minimizer $\hat{w}^*$. **(c)** SGD tends to stay longer on the flat side of asymmetric valleys, therefore SGD averaging automatically produces a bias towards the flat side.

and small batch training are shown to be connected by a path without any "bumps" [43]. In other words, a "sharp minimum" and a "flat minimum" may in fact belong to a same minimum in high dimensional space. Therefore, the notion of flat and sharp minima seems to be an oversimplification of the empirical loss surface.

In this paper, we expand the notion of flat and sharp minima by introducing the concept of *asymmetric valleys*. We observe that the loss surfaces of many neural networks are locally asymmetric. In specific, there exist many directions such that the loss increases abruptly along one side, and grows rather slowly along the opposite side (see Figure 1(b) as an illustration). We formally define this kind of local minima as asymmetric valleys. As we will show in Section 6, asymmetric valleys generate interesting illusions in high dimensional space. For example, located in the same valley shown in Figure 1(b), $\tilde{w}$ may appear to be a wider and flatter minimum than $\hat{w}$ as the former is farther away from the sharp side.

Asymmetric valleys also introduce novel insights to generalization. Folklore says when the exact minimizer is flat, it tends to generalize better as it is more stable with respect to loss surface perturbations [32]. Instead of following this argument, we show that in asymmetric valleys, the solution biased towards the flat side of the valley generalizes better than the exact minimizer, under mild assumptions. This result has at least two interesting implications: (1) converging to *which* local minimum (if there are many) may not be critical for modern deep networks. However, it matters a lot *where* the solution locates; and (2) the solution with lowest *a priori* generalization error is not necessarily the minimizer of the training loss.

Given that a biased solution is preferred for asymmetric valleys, an immediate question is how we can find such solutions in practice. It turns out that simply averaging the weights along the SGD trajectory, naturally leads to the desired solutions. We give a theoretical analysis to support this argument, see Figure 1(c) for an illustration. Our result nicely complements a series of recent empirical observations, which demonstrated that averaged SGD has better performance over plain SGD, for various scenarios including supervised/unsupervised/low-precision training [25, 5, 51].

In addition, we provide empirical analysis to verify our theoretical results and support our claims. For example, we show that asymmetric valleys are indeed prevalent in modern deep networks, and solutions with lower generalization error has bias towards the flat side of the valley.

## 2 Related Work

**Neural network landscape**. Neural network landscape analysis is an active and exciting area [16, 34, 15, 40, 49, 10, 43]. For example, [12, 13] observed that essentially all local minima are connected together with simple paths. In [22], cyclic learning rate was used to explore multiple local optima along the training trajectory for model ensembling. There are also appealing visualizations for the neural network landscape [34].

**Sharp and flat minima**. The discussion of sharp and flat local minima dates back to [20], and recently regains its popularity. For example, Keskar et al. [32] proposed that large batch SGD finds

sharp minima, which leads to poor generalization. In [8], an entropy regularized SGD was introduced to explicitly searching for flat minima. It was later pointed out that large batch SGD can yield comparable performance when the learning rate or the number of training iterations are properly set [21, 17, 47, 35, 46, 26]. Moreover, [11] showed that from a given flat minimum, one could construct another minimum with arbitrarily sharp directions but equally good performance. In this paper, we argue that the description of sharp or flat minima is an oversimplification. There may simultaneously exist steep directions, flat directions, and asymmetric directions for the same minimum.

**SGD optimization and generalization**. As the de facto optimization tool for deep networks, SGD and its variants are extensively studied in the literature. For example, it is shown that they could escape saddle points or sharp local minima under reasonable assumptions [14, 28–30, 50, 1–3, 33]. For convex functions [41] or strongly convex but non-smooth functions [42], SGD averaging is shown to give better convergence rate. In addition, it can also achieve higher generalization performance for Lipschitz functions in theory [44, 7], or for deep networks in practice [22, 25, 5, 51]. Discussions on the generalization bound of neural networks can be found in [6, 39, 37, 31, 38, 4, 52]. We show that SGD averaging has implicit bias on the flat sides of the minima. Previously, it was shown that SGD has other kinds of implicit bias as well [48, 27, 18].

## 3 Asymmetric Valleys

In this section, we give a formal definition of asymmetric valley, and empirically show that it is prevalent in the loss landscape of modern deep neural networks.

**Preliminaries.** In supervised learning, we seek to optimize $\boldsymbol{w}^* \triangleq \arg\min_{\boldsymbol{w} \in \mathbb{R}^d} \mathsf{L}(\boldsymbol{w})$, where $\mathsf{L}(\boldsymbol{w}) \triangleq \mathbb{E}_{\boldsymbol{x} \sim \mathcal{D}}[f(\boldsymbol{x}; \boldsymbol{w})] \in \mathbb{R}^d \rightarrow \mathbb{R}$ is the population loss, $\boldsymbol{x} \in \mathbb{R}^m$ is the input sampled from distribution $\mathcal{D}$, $\boldsymbol{w} \in \mathbb{R}^d$ denotes the model parameter, and $f \in \mathbb{R}^m \times \mathbb{R}^d \rightarrow \mathbb{R}$ is the loss function. Since the data distribution $\mathcal{D}$ is usually unknown, instead of optimizing $\mathsf{L}$ directly, we often use SGD to find the empirical risk minimizer $\hat{\boldsymbol{w}}^*$ for a set of random samples $\{\boldsymbol{x}_i\}_{i=1}^n$ from $\mathcal{D}$ (a.k.a. training set): $\hat{\boldsymbol{w}}^* \triangleq \arg\min_{\boldsymbol{w} \in \mathbb{R}^d} \hat{\mathsf{L}}(\boldsymbol{w})$, where $\hat{\mathsf{L}}(\boldsymbol{w}) \triangleq \frac{1}{n} \sum_{i=1}^n f(\boldsymbol{x}_i; \boldsymbol{w})$.

In practice, it is numerically infeasible to find or test the exact local minimizer $\hat{\boldsymbol{w}}^*$. Fortunately, our theoretical results only depend on a good enough solution rather than an exact local minimum, as we will formally define in Section 4. For simplicity, we still refer to such solutions as "local minima", although our analysis generalizes to "solutions found by SGD".

### 3.1 Definition of asymmetric valley

Before formally introducing asymmetric valleys, we first define asymmetric directions.

**Definition 1** (Asymmetric direction). *Given constants $p > 0, \overline{r} > \underline{r} > 0, c > 1$, a direction $\boldsymbol{u}$ is $(\overline{r}, \underline{r}, p, c)$-asymmetric with respect to point $\boldsymbol{w} \in \mathbb{R}^d$ and loss function $\hat{\mathsf{L}}$, if $\nabla_l \hat{\mathsf{L}}(\boldsymbol{w} + l\boldsymbol{u}) < p$, and $\nabla_l \hat{\mathsf{L}}(\boldsymbol{w} - l\boldsymbol{u}) > cp$ for $l \in (\underline{r}, \overline{r})$.*

In the above definition, $\boldsymbol{u} \in \mathbb{R}^d$ is a unit vector representing a direction such that the points on this direction passing $\boldsymbol{w} \in \mathbb{R}^d$ can be written as $\boldsymbol{w} + l\boldsymbol{u}$ for $l \in (-\infty, \infty)$. Intuitively, the loss landscape in the interval $(-\overline{r}, -\underline{r})$ is "sharp", while it is "flat" in the region $(\underline{r}, \overline{r})$. Note that we purposely leave out the region $(-\underline{r}, \underline{r})$ without making further assumptions on it to comply with the fact that the second order derivatives of the loss function is usually continuous. It is impractical to assume the slope of the loss function change abruptly at the point $l = 0$.

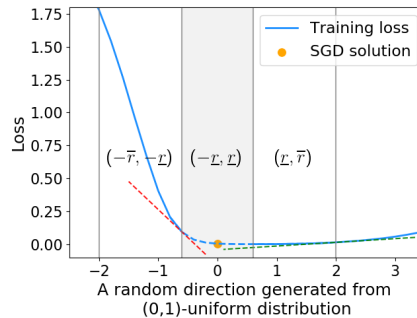

Figure 2: An asymmetric direction of a solution on the loss landscape of ResNet-110 trained on CIFAR-10.

As a concrete example, Figure 2 shows an asymmetric direction for a local minimum in ResNet-110 trained on the CIFAR-10 dataset. We verified that it is a $(2.0, 0.6, 0.03, 15)$-asymmetric

direction, which means in the region $(-2.0, -0.6) \cup (0.6, 2.0)$ the gradients are asymmetric with a relative ratio of $c = 15$.

With this Definition 1, we now formally define the *asymmetric valley*[2].

**Definition 2** (Asymmetric valley). *Given constants $p, \overline{r} > \underline{r} > 0, c > 1$, a solution $\hat{\boldsymbol{w}}^*$ of $\hat{\mathsf{L}} \in \mathbb{R}^d \to \mathbb{R}$ is a $(\overline{r}, \underline{r}, p, c)$-asymmetric valley, if there exists at least one direction $\boldsymbol{u}$ such that $\boldsymbol{u}$ is $(\overline{r}, \underline{r}, p, c)$-asymmetric with respect to $\hat{\boldsymbol{w}}^*$ and $\hat{\mathsf{L}}$.*

### 3.2 Asymmetric valleys in neural networks

Empirically, by taking random directions with value $(0, 1)$ in each dimension, we can find an asymmetric direction for a given solution $\boldsymbol{w}^*$ with decent probability. We perform experiments with widely used deep networks, i.e.,ResNet-56, ResNet-110, ResNet-164 [19], VGG-16 [45] and DenseNet-100 [23], on the CIFAR-10, CIFAR-100, SVHN and STL-10 image classification datasets. For each model on each dataset, we conduct 5 independent runs. The results show that we can *always* find asymmetric directions with certain specification $(\overline{r}, \underline{r}, p, c)$ with $c > 2$, which means all the solutions that SGD found are located in asymmetric valleys. Asymmetric valleys widely exist in both simple and complex models, see Appendix A, Appendix E and Appendix F. For example, in Appendix A we show that asymmetric valleys exist in a simple 2 layer network with only 2 parameters.

## 4 Bias and Generalization

As we show in the previous section, in the context of deep learning most local minima in practice are *asymmetric*, i.e., they might be sharp on one direction, but flat on the opposite direction. Therefore, it is interesting to investigate the generalization ability of a solution $\boldsymbol{w}$ in this scenario, which may lead to different results as those obtained under the common symmetric assumption. In this section, we prove that a *biased* solution on the flat side of an asymmetric valley yields lower generalization error than the exact empirical minimizer $\hat{\boldsymbol{w}}^*$ in that valley.

### 4.1 Theoretical analysis

Before presenting our theorem, we first introduce two mild assumptions. We will show that they empirically hold on modern deep networks in Section 4.2.

The first assumption (Assumption 1) states that there exists a shift between the empirical loss and true population loss. This is a common assumption in the previous works, e.g., [32], but was usually presented in an informal way. Here we define the "shift" in formally. Without loss of generality, we will compare the empirical loss $\hat{\mathsf{L}}$ with $\mathsf{L}' \triangleq \mathsf{L} - \min_{\boldsymbol{w}} \mathsf{L}(\boldsymbol{w}) + \min_{\boldsymbol{w}} \hat{\mathsf{L}}(\boldsymbol{w})$ to remove the "vertical difference" between $\hat{\mathsf{L}}$ and $\mathsf{L}$. Notice that $\min_{\boldsymbol{w}} \mathsf{L}(\boldsymbol{w})$ and $\min_{\boldsymbol{w}} \hat{\mathsf{L}}(\boldsymbol{w})$ are constants and do not affect our generalization guarantee.

**Definition 3** $((\boldsymbol{\delta}, R)$-shift gap). *For $\xi \geq 0$, $\boldsymbol{\delta} \in \mathbb{R}^d$, and fixed functions $\mathsf{L}$ and $\hat{\mathsf{L}}$, we define the $(\boldsymbol{\delta}, R)$-shift gap between $\mathsf{L}$ and $\hat{\mathsf{L}}$ with respect to a point $\boldsymbol{w}$ as*

$$\xi_{\boldsymbol{\delta}}(\boldsymbol{w}) = \max_{\boldsymbol{v} \in \mathbb{B}(R)} |\mathsf{L}'(\boldsymbol{w} + \boldsymbol{v} + \boldsymbol{\delta}) - \hat{\mathsf{L}}(\boldsymbol{w} + \boldsymbol{v})|$$

*where $\mathsf{L}'(\boldsymbol{w}) \triangleq \mathsf{L}(\boldsymbol{w}) - \min_{\boldsymbol{w}} \mathsf{L}(\boldsymbol{w}) + \min_{\boldsymbol{w}} \hat{\mathsf{L}}(\boldsymbol{w})$, and $\mathbb{B}(R)$ is the d-dimensional ball with radius $R$ centered at $\mathbf{0}$.*

From the above definition, we know that the two functions match well after the shift $\boldsymbol{\delta}$ if $\xi_{\boldsymbol{\delta}}(\boldsymbol{w})$ is very small. For example, $\xi_{\boldsymbol{\delta}}(\boldsymbol{w}) = 0$ means $\mathsf{L}$ is locally identical to $\hat{\mathsf{L}}$ after the shift $\boldsymbol{\delta}$. Since $\hat{\mathsf{L}}$ is computed on a set of random samples from $\mathcal{D}$, the actual shift $\boldsymbol{\delta}$ between $\hat{\mathsf{L}}$ and $\mathsf{L}$ is a random variable, ideally with zero expectation[3].

**Assumption 1** (Random shift assumption). *For a given population loss $\mathsf{L}$ and a random empirical loss $\hat{\mathsf{L}}$, constants $R > 0, \bar{r} \geq \underline{r} > 0, \xi \geq 0$, a vector $\bar{\boldsymbol{\delta}} \in \mathbb{R}^d$ with $\bar{r} \geq \bar{\boldsymbol{\delta}}_i \geq \underline{r}$ for all $i \in [d]$, a minimizer $\hat{\boldsymbol{w}}^*$, we assume that there exists a random variable $\boldsymbol{\delta} \in \mathbb{R}^d$ correlated with $\hat{\mathsf{L}}$ such that $\Pr(\boldsymbol{\delta}_i = \bar{\boldsymbol{\delta}}_i) = \Pr(\boldsymbol{\delta}_i = -\bar{\boldsymbol{\delta}}_i) = \frac{1}{2}$ for all $i \in [d]$, and the $(\boldsymbol{\delta}, R)$-shift gap between $\mathsf{L}$ and $\hat{\mathsf{L}}$ with respect to $\hat{\boldsymbol{w}}^*$ is bounded by $\xi$.*

Clearly, $\boldsymbol{\delta}$ has $2^d$ possible values for a given shift vector $\bar{\boldsymbol{\delta}}$, each with probability $2^{-d}$. Notice that Assumption 1 does not say that the difference between $\mathsf{L}$ and $\hat{\mathsf{L}}$ can only be one of the $2^d$ possible $\boldsymbol{\delta}$. Instead, it says after applying the shift $\boldsymbol{\delta}$, the two functions have bounded $L_\infty$ distance, which is a much milder assumption. It is also worth noting that our Definition 1 can mask out the central interval $(-\underline{r}, \underline{r})$ because we have $\bar{r} \geq \bar{\boldsymbol{\delta}}_i \geq \underline{r}$ in Assumption 1. Therefore, $\underline{r}$ cannot be arbitrarily large, otherwise Assumption 1 does not hold. Our second assumption stated below can be seen as an extension of Definition 2.

**Assumption 2** (Locally asymmetric). *For a given population loss $\hat{\mathsf{L}}$, and a minimizer $\hat{\boldsymbol{w}}^*$, there exist orthogonal directions $\boldsymbol{u}^1, \cdots, \boldsymbol{u}^k \in \mathbb{R}^d$ s.t. $\boldsymbol{u}^i$ is $(\bar{r}, \underline{r}, p_i, c_i)$-asymmetric with respect to $\hat{\boldsymbol{w}}^* + \boldsymbol{v} - \langle \boldsymbol{v}, \boldsymbol{u}^i \rangle \boldsymbol{u}^i$ for all $\boldsymbol{v} \in \mathbb{B}(R')$ and $i \in [k]$.*

Assumption 2 states that if $\boldsymbol{u}^i$ is an asymmetric direction at $\hat{\boldsymbol{w}}^*$, then the point $\hat{\boldsymbol{w}}^* + \boldsymbol{v} - \langle \boldsymbol{v}, \boldsymbol{u}^i \rangle \boldsymbol{u}^i$ that deviates from $\hat{\boldsymbol{w}}^*$ along the perpendicular direction of $\boldsymbol{u}^i$, is also asymmetric along the direction of $\boldsymbol{u}^i$. In other words, the *neighborhood* around $\hat{\boldsymbol{w}}^*$ is an asymmetric valley.

Under the above assumptions, we are ready to state our theorem, which says the empirical minimizer is not necessarily the optimal solution, and a biased solution leads to better generalization. We defer the proof to Appendix B.

**Theorem 1** (Bias leads to better generalization). *For any $\boldsymbol{l} \in \mathbb{R}^k$, if Assumption 1 holds for $R = \|\boldsymbol{l}\|_2$, Assumption 2 holds for $R' = \|\bar{\boldsymbol{\delta}}\|_2 + \|\boldsymbol{l}\|_2$, and $\frac{4\xi}{(c_i-1)p_i} < l_i \leq \max\{\bar{r} - \bar{\boldsymbol{\delta}}_i, \bar{\boldsymbol{\delta}}_i - \underline{r}\}$, then we have*

$$\mathbb{E}_{\boldsymbol{\delta}} \mathsf{L}(\hat{\boldsymbol{w}}^*) - \mathbb{E}_{\boldsymbol{\delta}} \mathsf{L}\left(\hat{\boldsymbol{w}}^* + \sum_{i=1}^k l_i \boldsymbol{u}^i\right) \geq \sum_{i=1}^k (c_i - 1) l_i p_i / 2 - 2k\xi > 0$$

**Remark on Theorem 1.** It is widely known that the empirical minimizer is usually different from the true optimum. However, in practice it is difficult to know how the training loss shifts from the population loss. Therefore, the best we could it to minimize the empirical loss function (with some regularizers). However, Theorem 1 states that in the asymmetric case, we should pick a biased solution even if the shift is unknown. This insight can be distilled into practical algorithms to achieve better generalization, as we will discuss in Section 5.

## 4.2 Validating assumptions

We conducted a series of experiments with modern deep networks to show that the two assumptions introduced above are generally valid.

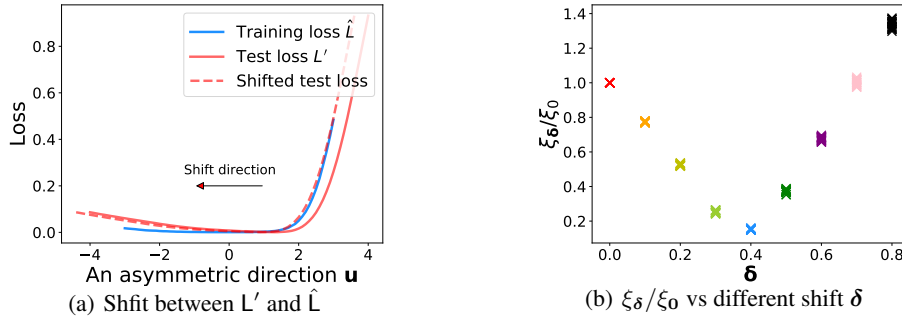

(a) Shfit between $\mathsf{L}'$ and $\hat{\mathsf{L}}$      (b) $\xi_{\boldsymbol{\delta}}/\xi_{\boldsymbol{0}}$ vs different shift $\boldsymbol{\delta}$

Figure 3: Shift exists between empirical loss and population loss for ResNet-110 on CIFAR-10.

**Verification of Assumption 1.** We show that a shift between $\mathsf{L}$ and $\hat{\mathsf{L}}$ is quite common in practice, by taking a ResNet-110 trained on CIFAR-10 as an example. Notice that we use test loss to represent

L in practice. Since we could not visualize a shift in a high dimensional space, we randomly sample an asymmetric direction $\boldsymbol{u}$ (more results are shown Appendix C) at the SGD solution $\hat{\boldsymbol{w}}^*$. The blue and red curves shown in Figure 3(a) are obtained by calculating $\hat{\mathsf{L}}(\hat{\boldsymbol{w}}^* + l\boldsymbol{u})$ and $\mathsf{L}'(\hat{\boldsymbol{w}}^* + l\boldsymbol{u})$ for $l \in [-3, 3]$, which correspond to the training and test loss, respectively.

We then try different shift values of $\boldsymbol{\delta}$ to "match" the two curves. As shown in Figure 3(a), after applying a horizontal shift $\boldsymbol{\delta} = 0.4$ to the test loss, the two curves overlap almost perfectly. Quantitatively, we can use the *shift gap* defined in Definition 3 to evaluate how well the two curves match each other after shifting. It turns out that $\xi_{\boldsymbol{\delta}=0.4} = 0.03$, which is much lower than $\xi_{\boldsymbol{\delta}=0} = 0.22$ before shifting ($\boldsymbol{\delta}$ has only one dimension here). In Figure 3(b), we plot $\xi_{\boldsymbol{\delta}}/\xi_{\boldsymbol{0}}$ as a function of $\boldsymbol{\delta}$. Clearly, there exists a $\boldsymbol{\delta}$ that minimizes this ratio, indicating a good match.

We conducted the same experiments for different directions, models and datasets, and similar observations were made. Please refer to Appendix C for more results.

**Verification of Assumption 2.** This is a mild assumption that can be verified empirically. For example, we take a SGD solution of ResNet-110 on CIFAR-10 as $\hat{\boldsymbol{w}}^*$, and specify an asymmetric direction $\boldsymbol{u}$ for $\hat{\boldsymbol{w}}^*$. We then randomly sample 100 different local adjustments for $\boldsymbol{v} \in \mathbb{B}(25)$. Based on these adjustments, we present the mean loss curves and standard variance zone on the asymmetric direction $\boldsymbol{u}$ for all the points $\hat{\boldsymbol{w}}^* + \boldsymbol{v} - \langle \boldsymbol{v}, \boldsymbol{u} \rangle \boldsymbol{u}$ in Figure 4. As we can see, the variance of these curves are very small, which means all of them are similar to each other. Moreover, we verified that $\boldsymbol{u}$ is $(4, 2, 0.1, 5.22)$-asymmetric with respect to all neighboring points.

# 5 Averaging Generates Good Bias

In the previous section, we show that when the loss landscape of a local minimum is asymmetric, a solution with bias towards the flat side of the valley has better generalization performance. One immediate question is that how can we obtain such a solution via practical algorithms? Below we show that it can be achieved by simply taking the average of SGD iterates during the course of training. We first analyze the one dimensional case in Section 5.1, and then extend the analysis to the high dimensional case in Section 5.2.

Note that weight averaging is a classical algorithm in optimization [41], and recently regained its popularity in the context of deep learning [25, 5, 51]. Our following analysis can be viewed as a theoretical justification of recent algorithms that based on SGD iterates averaging.

## 5.1 One dimensional case

For asymmetric functions, as long as the learning rate is not too small, SGD will oscillate between the flat side and the sharp side. Below we focus on one round of oscillation, and show that the average of the iterates in each round has a bias on the flat side. Consequently, by aggregating all rounds of oscillation, averaging SGD iterates leads to a bias as well.

For each individual round $i$, we assume that it starts from the iteration when SGD goes from sharp side to flat side (denoted as $w_0^i$), and ends at the iteration exactly before the iteration that SGD goes from sharp side to flat side again (denoted as $w_{T_i}^i$). Here $T_i$ denotes the number of iterations in the $i$-th rounds. The average iterate in the $i$-th round can be written as $\bar{w} \triangleq \frac{1}{T_i} \sum_{j=0}^{T_i} w_j^i$. For notational simplicity, we will omit the super script $i$ on $w_j^i$.

The following theorem shows that the expectation of the average has bias on the flat side. To get a formal lower bound on $\bar{w}$, we consider the asymmetric case where $\underline{r} = 0$, and also assume lower bounds for the gradients on the function. We defer the proof to Appendix D.

**Theorem 2** (SGD averaging generates a bias). *Assume that a local minimizer $w^* = 0$ is a $(r, 0, a_+, c)$-asymmetric valley, where $b_- \leq \nabla\mathsf{L}(w) \leq a_- < 0$ for $w < 0$, and $0 < b_+ \leq \nabla\mathsf{L}(w) \leq a_+$ for $w \geq 0$. Assume $-a_- = ca_+$ for a large constant c, and $\frac{-(b_- - \nu)}{b_+} = c' < \frac{e^{c/3}}{6}$. The SGD updating rule is $w_{t+1} = w_t - \eta(\nabla L(w) + \omega_t)$ where $\omega_t$ is the noise and $|\omega_t| < \nu$, and assume $\nu \leq a_+$. Then we have*

$$\mathbb{E}[\bar{w}] > c_0 > 0,$$

*where $c_0$ is a constant that only depends on $\eta, a_+, a_-, b_+, b_-$ and $\nu$.*

Theorem 2 can be intuitively explained by Figure 5. If we run SGD on this one dimensional function, it will stay at the flat side for more iterations as the magnitude of the gradient on this side is much smaller. Therefore, the average of the locations is biased towards the flat side.

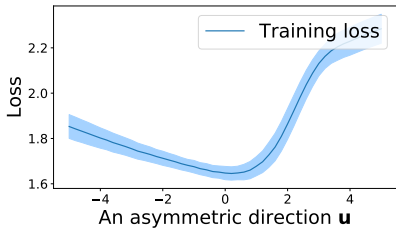

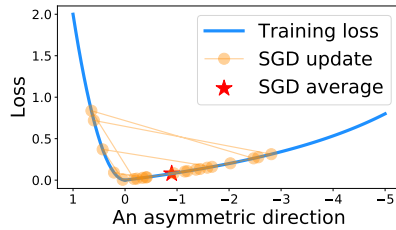

Figure 4: Training loss mean and variance for the neighborhood of $\hat{w}^*$ at the direction of $u$.

Figure 5: SGD iterates and their average on an asymmetric function.

## 5.2 High dimensional case

For high dimensional functions, the analysis on averaging SGD iterates would be more complicated compared to that given in the previous subsection. However, if we only care about the bias on a specific direction $u$, we could directly apply Theorem 2 with one additional assumption. Specifically, if the projections of the loss function onto $u$ along the SGD trajectory satisfy the assumptions in Theorem 2, i.e., being asymmetric and the gradient on both sides have upper and lower bounds, then the claim of Theorem 2 directly applies. This is because only the gradient along the direction $u$ will affect the SGD trajectory projected onto $u$, and we could safely omit all other directions.

We find that this assumption holds empirically. For a given SGD solution, we fix a random asymmetric direction $u \in \mathbb{R}^d$, and sample the loss surface on direction $u$ that passes the $t$-th epoch of SGD trajectory (denoted as $w_t$), i.e., evaluate $\hat{\mathsf{L}}(w_t + lu)$, for $0 \le t \le 200$ and $l \in [-15, 15]$. As shown in the Figure 6, after the first $40$ epochs, the projected loss surfaces becomes relatively stable. Therefore, we can directly apply Theorem 2 to the direction $u$.

As we will see in Section 6.1, compared with SGD solutions, SGD averaging indeed creates bias along different asymmetric directions, as predicted by our theory.

# 6 Experimental Observations

In this section, we empirically show that asymmetric valleys create interesting illusions when visualizing high dimensional loss landscape in low dimensional space. In addition, as a refinement of judging the generalization performance by the sharpness/flatness of a local minimum, we show that *where* the solution locates at a local minimum basin is important. We also find that batch normalization [24] seems to be a major cause for asymmetric valleys in deep networks, but the results are deferred to Appendix H due to space limit.

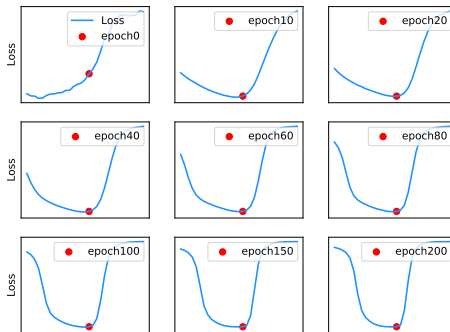

Figure 6: Projection of the training loss surface onto an asymmetric direction $u$

## 6.1 Experiments with weight averaging

Recently, Izmailov et al. [25] proposed the stochastic weight averaging (SWA) algorithm, which explicitly takes the average of SGD iterates to achieve better generalization. Inspired by their observation that "SWA leads to solutions corresponding to wider optima than SGD", we provide a more refined explanation in this subsection. That is, averaging weights leads to "biased" solutions in an asymmetric valley, which correspond to better generalization.

Specifically, we run the SWA algorithm (with deceasing learning rate) with popular deep networks, including ResNet-56, ResNet-110, ResNet-164, VGG-16, and DenseNet-100, on various datasets

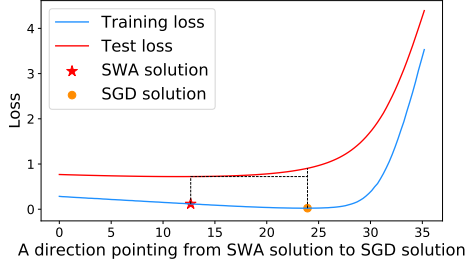
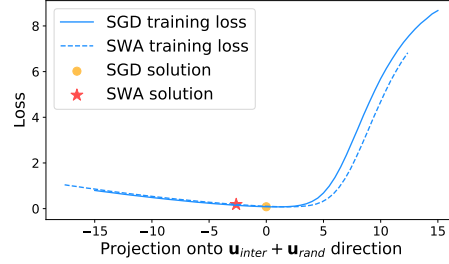

Figure 7: SWA solution and SGD solution interpolation (ResNet-164 on CIFAR-100)

Figure 8: The average of SGD has a bias on flat side (ResNet-110 on CIFAR-100)

Table 1: Training and test accuracy on CIFAR-100.

| Network | CIFAR-100 | |
|---|---|---|
| | train | test |
| ResNet-110-SWA | 94.98% | 78.94% |
| ResNet-110-SGD | 97.52% | 78.29% |
| ResNet-164-SWA | 97.48% | 80.69% |
| ResNet-164-SGD | 99.12% | 76.56% |

including CIFAR-10, CIFAR-100, SVHN and STL-10, following the configurations in [25]. Then we run SGD with small learning rate *from the SWA solutions* to find a solution located in the same basin (denoted as SGD).

In Figure 7, We draw an interpolation between the solutions obtained by SWA and SGD[4]. One can observe that there is no "bump" between these two solutions, meaning they are located in the same basin. Clearly, the SWA solution is biased towards the flat side, which verifies our theoretical analysis in Section 5. Further, we notice that although the biased SWA solution has higher training loss than the solution found by SGD, it indeed yields lower test loss. This verifies our analysis in Section 4. Similar observations are made on other networks and other datasets, which we present in Appendix E.

To further support our claim, we list our result in Table 1, from which we can observe that SGD solutions always have higher training accuracy, but worse test accuracy, compared to SWA solutions. This supports our claim in Theorem 1, which states that a bias towards the flat sides of asymmetric valleys could help improve generalization, although it yields higher training error.

**Verifying Theorem 2.** We further verify that averaging SGD solutions creates a bias towards the flat side in expectation for many other asymmetric directions, not just for the specific direction we discussed above.

We take a ResNet-110 trained on CIFAR-100 as an example. Denote $u_{inter}$ as the unit vector pointing from the SGD solution to the SWA solution, $u_{rand}$ as another unit random direction, and the direction $u_{inter} + u_{rand}$ is used to explore the asymmetric landscape.

The results are shown in Figure 8, from which we can observe that SWA has a bias on the flat side compared with the SGD solution. We create 10 different random vectors for each network and each dataset, and similar observations can be made (see more examples in Appendix F).

**Batch size effect** In addition to SWA algorithm, we also observe similar trend when training with different batch sizes. The results are deferred to Appendix G.

## 6.2 Illusions created by asymmetric valleys

We further point out that visualizing the "width" of a given solution $w$ in a low-dimensional space may lead to illusive results. For example, one visualization technique used in [25] is to show how the loss changes along many random directions $v_i$'s drawn from the $d$-dimensional Gaussian distribution.

We take the large batch and small batch solutions from the previous subsection as an example. Figure 9 visualizes the "width" of the two solutions using the method described above. From the

figure, one may draw the conclusion that small batch training leads to a wider minimum compared to large batch training. However, these two solutions are in fact from the *same* basin (see the discussion in Appendix G). In other words, the loss curvature near the two solutions looks different because they are located at *different locations* in a same asymmetric valley, instead of being located at *different local minima*. Similar observation holds for SWA and SGD solutions, see Figure 10[5] .

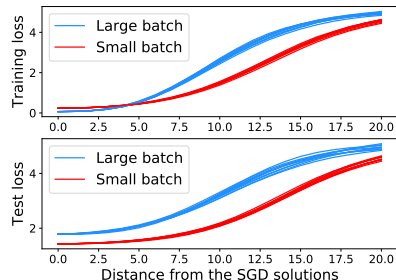

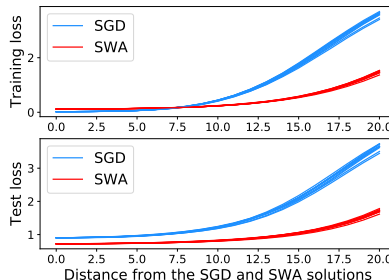

Figure 9: Random ray of large batch and small batch solution.

Figure 10: Random ray of SGD and SWA solution

# 7   Conclusion

In this paper, we introduced the notion of asymmetric valley to characterize the loss landscape of deep networks, expanding the current research that simply categorizes local minima by sharpness/flatness. This notion allowed us to analyze and understand the geometry of loss landscape from a new perspective. For example, based on a formal definition of asymmetric valley, we showed that a biased solution lying on the flat side of the valley generalizes better than the exact empirical minimizer. Further, it is proved that by averaging the weights obtained along the SGD trajectory naturally leads to such biased solution. We also conducted extensive experiments with state-of-the-art deep models to analyze the properties of asymmetric valleys. It is showed that due to the existence of asymmetric valleys, intriguing illustions can be created when visualizing high dimensional loss surface in the 1D space. We hope this work will deepen our understanding on the loss landscape of deep neural networks, and inspire new theories and algorithms that further improve generalization.

# Acknowledgment

This work has been supported in part by the Zhongguancun Haihua Institute for Frontier Information Technology. Gao Huang is supported in part by Beijing Academy of Artificial Intelligence (BAAI) under grant BAAI2019QN0106.

## Footnotes

[2]Here we abuse the name "valley", since $\hat{\boldsymbol{w}}^*$ is essentially a point at the center of a valley.

[3]It may not be zero, as we are talking about the shift between two loss functions, rather than the difference between empirical/population loss values.

[4]Izmailov et al. [25] have done a similar experiment.

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
