[Supplementary Material · neurips_2019.pdf]

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

# A    Additional Figures for Section 3.2: Asymmetric Directions

To show that asymmetric valley can be commonly observed, we conduct experiments ranging from the simplest network to modern deep neural networks.

**A simple case** First, we will show that asymmetric valley can be observed on a simple MLP (one hidden layer with 10 hidden neurons) on a logistic regression task in Figure 11

Figure 11: Asymmetric direction for a solution of MLP on logistic regression. $(\overline{r}, \underline{r}, p, c) = (10.0, 5.0, 0.11, 6.0)$.

Figure 12: Asymmetric direction for a solution of ResNet-164 on CIFAR-10. $(\overline{r}, \underline{r}, p, c) = (4.0, 2.5, 0.033, 4.8)$.

**Other datasets and networks** See Figure 12, Figure 13, Figure 14, Figure 15, and Figure 16.

Figure 13: Asymmetric direction for a solution of DenseNet-100 on CIFAR-10. $(\overline{r}, \underline{r}, p, c) = (7.0, 5.0, 0.030, 4.8)$.

Figure 14: Asymmetric direction for a solution of ResNet-110 on CIFAR-100. $(\overline{r}, \underline{r}, p, c) = (7.0, 5.0, 0.039, 2.7)$.

Figure 15: Asymmetric direction for a solution of ResNet-164 on CIFAR-100. $(\overline{r}, \underline{r}, p, c) = (7.0, 5.0, 0.031, 2.5)$.

Figure 16: Asymmetric direction for a solution of DenseNet-100 on CIFAR-100. $(\overline{r}, \underline{r}, p, c) = (8.5, 6.5, 0.087, 2.1)$.

# B    Proof for Theorem 1

*Proof.* Since $\boldsymbol{\delta}$ has $2^d$ possible value for a given $\overline{\boldsymbol{\delta}}$, we can use an integer $j \in \{0, \cdots, 2^d - 1\}$ to represent each value. When writing $j$ in binary, its $i$-th digit represents whether $\boldsymbol{\delta}_i = \overline{\boldsymbol{\delta}}_i$ (equal to 1) or $\boldsymbol{\delta}_i = -\overline{\boldsymbol{\delta}}_i$ (equal to 0). We use $j \wedge 2^i$ to represent the bitwise AND operator between $j$ and $2^i$, which equals $0$ if the $i$-th digit of $j$ is 0.

To prove our theorem, it suffices to show that for any $i \in [k]$,

$$\mathbb{E}_{\boldsymbol{\delta}} \mathsf{L} \left( \hat{\boldsymbol{w}}^* + \sum_{i_0=1}^{i-1} \boldsymbol{l}_{i_0} \boldsymbol{u}_{i_0} \right) - \mathbb{E}_{\boldsymbol{\delta}} \mathsf{L} \left( \hat{\boldsymbol{w}}^* + \sum_{i_0=1}^{i} \boldsymbol{l}_{i_0} \boldsymbol{u}_{i_0} \right) \geq (c_i - 1) l_i p_i / 2 - 2\xi > 0 \qquad (1)$$

If (1) is true, it suffices to take summation over $i$ on both sides, and we will get our conclusion. Therefore, below we will prove (1).

$$\mathbb{E}_{\boldsymbol{\delta}} \mathsf{L} \left( \hat{\boldsymbol{w}}^* + \sum_{i_0=1}^{i-1} \boldsymbol{l}_{i_0} \boldsymbol{u}_{i_0} \right) - \min_{\boldsymbol{w}} \mathsf{L}(\boldsymbol{w}) + \min_{\boldsymbol{w}} \hat{\mathsf{L}}(\boldsymbol{w})$$

$$= \mathbb{E}_{\boldsymbol{\delta}} \mathsf{L}' \left( \hat{\boldsymbol{w}}^* + \sum_{i_0=1}^{i-1} \boldsymbol{l}_{i_0} \boldsymbol{u}_{i_0} \right) \overset{①}{\geq} \frac{1}{2^d} \sum_{j=0}^{2^d-1} \hat{\mathsf{L}} \left( \hat{\boldsymbol{w}}^* + \sum_{i_0=1}^{i-1} \boldsymbol{l}_{i_0} \boldsymbol{u}_{i_0} + \boldsymbol{\delta}^j \right) - \xi$$

$$= \frac{1}{2^d} \sum_{\substack{j=0 \\ j \wedge 2^i = 0}}^{2^d-1} \left[ \hat{\mathsf{L}} \left( \hat{\boldsymbol{w}}^* + \sum_{i_0=1}^{i-1} \boldsymbol{l}_{i_0} \boldsymbol{u}_{i_0} + \boldsymbol{\delta}^j \right) + \hat{\mathsf{L}} \left( \hat{\boldsymbol{w}}^* + \sum_{i_0=1}^{i-1} \boldsymbol{l}_{i_0} \boldsymbol{u}_{i_0} + \boldsymbol{\delta}^{j+2^i} \right) \right] - \xi \qquad (2)$$

Where ① holds by Assumption 1, and the fact that $\| \sum_{i_0=1}^{i-1} \boldsymbol{l}_{i_0} \boldsymbol{u}_{i_0} \|_2 \leq \| \boldsymbol{l} \|_2 = R$. For every $j$ s.t. $j \wedge 2^i = 0$,

$$\hat{\boldsymbol{w}}^* + \sum_{i_0=1}^{i} \boldsymbol{l}_{i_0} \boldsymbol{u}_{i_0} + \boldsymbol{\delta}^j$$

$$= \hat{\boldsymbol{w}}^* + \sum_{i_0=1}^{i} \boldsymbol{l}_{i_0} \boldsymbol{u}_{i_0} + \boldsymbol{\delta}^j + \langle \boldsymbol{\delta}^j, \boldsymbol{u}^i \rangle \boldsymbol{u}^i - \langle \boldsymbol{\delta}^j, \boldsymbol{u}^i \rangle \boldsymbol{u}^i$$

$$= \hat{\boldsymbol{w}}^* + \sum_{i_0=1}^{i-1} \boldsymbol{l}_{i_0} \boldsymbol{u}_{i_0} + \boldsymbol{\delta}^j - \bar{\boldsymbol{\delta}}_i \boldsymbol{u}^i - \langle \boldsymbol{\delta}^j, \boldsymbol{u}^i \rangle \boldsymbol{u}^i + l_i \boldsymbol{u}^i$$

$$= \hat{\boldsymbol{w}}^* + \sum_{i_0=1}^{i-1} \boldsymbol{l}_{i_0} \boldsymbol{u}_{i_0} + \boldsymbol{\delta}^j - \langle \boldsymbol{\delta}^j, \boldsymbol{u}^i \rangle \boldsymbol{u}^i + (l_i - \bar{\boldsymbol{\delta}}_i) \boldsymbol{u}^i$$

Since $\| \sum_{i_0=1}^{i-1} \boldsymbol{l}_{i_0} \boldsymbol{u}_{i_0} \|_2 \leq \| \boldsymbol{l} \|_2$, $\| \delta^j \|_2 = \| \bar{\boldsymbol{\delta}} \|_2$, we know that $\forall j$, $\sum_{i_0=1}^{i-1} \boldsymbol{l}_{i_0} \boldsymbol{u}_{i_0} + \boldsymbol{\delta}^j \in \mathbb{B}(R')$. By Assumption 2, for every $i \in [k]$, $\boldsymbol{u}^i$ is asymmetric with respect to $\hat{\boldsymbol{w}}^* + \sum_{i_0=1}^{i-1} \boldsymbol{l}_{i_0} \boldsymbol{u}_{i_0} + \boldsymbol{\delta}^j - \langle \boldsymbol{\delta}^j, \boldsymbol{u}^i \rangle \boldsymbol{u}^i$. Since $\boldsymbol{l}_i \leq \bar{\boldsymbol{\delta}}_i - \underline{r}$, we have $\boldsymbol{l}_i - \bar{\boldsymbol{\delta}}_i < -\underline{r}$. By the definition of asymmetric direction, we know

$$\hat{\mathsf{L}} \left( \hat{\boldsymbol{w}}^* + \sum_{i_0=1}^{i-1} \boldsymbol{l}_{i_0} \boldsymbol{u}_{i_0} + \boldsymbol{\delta}^j \right) \geq \hat{\mathsf{L}} \left( \hat{\boldsymbol{w}}^* + \sum_{i_0=1}^{i} \boldsymbol{l}_{i_0} \boldsymbol{u}_{i_0} + \boldsymbol{\delta}^j \right) + c_i l_i p_i \qquad (3)$$

Similarly,

$$\hat{\boldsymbol{w}}^* + \sum_{i_0=1}^{i} \boldsymbol{l}_{i_0} \boldsymbol{u}_{i_0} + \boldsymbol{\delta}^{j+2^i}$$

$$= \hat{\boldsymbol{w}}^* + \sum_{i_0=1}^{i-1} \boldsymbol{l}_{i_0} \boldsymbol{u}_{i_0} + \boldsymbol{\delta}^{j+2^i} + \langle \boldsymbol{\delta}^{j+2^i}, \boldsymbol{u}^i \rangle \boldsymbol{u}^i - \langle \boldsymbol{\delta}^{j+2^i}, \boldsymbol{u}^i \rangle \boldsymbol{u}^i + l_i \boldsymbol{u}^i$$

$$= \hat{\boldsymbol{w}}^* + \sum_{i_0=1}^{i-1} \boldsymbol{l}_{i_0} \boldsymbol{u}_{i_0} + \boldsymbol{\delta}^{j+2^i} - \langle \boldsymbol{\delta}^{j+2^i}, \boldsymbol{u}^i \rangle \boldsymbol{u}^i + (\bar{\boldsymbol{\delta}}_i + l_i) \boldsymbol{u}^i$$

Since $\boldsymbol{l}_i \leq r - \bar{\boldsymbol{\delta}}_i$, we have $\bar{\boldsymbol{\delta}}_i + l_i \leq r$. Therefore,

$$\hat{\mathsf{L}} \left( \hat{\boldsymbol{w}}^* + \sum_{i_0=1}^{i-1} \boldsymbol{l}_{i_0} \boldsymbol{u}_{i_0} + \boldsymbol{\delta}^{j+2^i} \right) \geq \hat{\mathsf{L}} \left( \hat{\boldsymbol{w}}^* + \sum_{i_0=1}^{i} \boldsymbol{l}_{i_0} \boldsymbol{u}_{i_0} + \boldsymbol{\delta}^{j+2^i} \right) - l_i p_i \qquad (4)$$

467 Combining (3) and (4), we have,

$$
(2) \geq \frac{1}{2^d} \sum_{\substack{j=0 \\ j \wedge 2^i = 0}}^{2^d-1} \left[ \hat{\mathsf{L}}\left(\hat{\boldsymbol{w}}^* + \sum_{i_0=1}^{i} \boldsymbol{l}_{i_0}\boldsymbol{u}_{i_0} + \boldsymbol{\delta}^j\right) + c_i \boldsymbol{l}_i p_i + \hat{\mathsf{L}}\left(\hat{\boldsymbol{w}}^* + \sum_{i_0=1}^{i} \boldsymbol{l}_{i_0}\boldsymbol{u}_{i_0} + \boldsymbol{\delta}^{j+2^i}\right) - \boldsymbol{l}_i p_i \right] - \xi
$$

$$
= \frac{1}{2^d} \sum_{j=0}^{2^d-1} \left[ \hat{\mathsf{L}}\left(\hat{\boldsymbol{w}}^* + \sum_{i_0=1}^{i} \boldsymbol{l}_i \boldsymbol{u}_{i_0} + \boldsymbol{\delta}^j\right) \right] + (c_i - 1)\boldsymbol{l}_i p_i / 2 - \xi
$$

$$
\overset{②}{\geq} \mathbb{E}_{\boldsymbol{\delta}} \mathsf{L}'\left(\hat{\boldsymbol{w}}^* + \sum_{i_0=1}^{i} \boldsymbol{l}_{i_0}\boldsymbol{u}_{i_0}\right) + (c_i - 1)\boldsymbol{l}_i p_i / 2 - 2\xi
$$

$$
= \mathbb{E}_{\boldsymbol{\delta}} \mathsf{L}\left(\hat{\boldsymbol{w}}^* + \sum_{i_0=1}^{i} \boldsymbol{l}_{i_0}\boldsymbol{u}_{i_0}\right) - \min_{\boldsymbol{w}} \mathsf{L}(\boldsymbol{w}) + \min_{\boldsymbol{w}} \hat{\mathsf{L}}(\boldsymbol{w}) + (c_i - 1)\boldsymbol{l}_i p_i / 2 - 2\xi
$$

468 Where ② holds by Assumption 1 and the fact that $\|\sum_{i_0=1}^{i} \boldsymbol{l}_{i_0}\boldsymbol{u}_{i_0}\|_2 \leq \|\boldsymbol{l}\|_2 = R$. That means,

$$
\mathbb{E}_{\boldsymbol{\delta}} \mathsf{L}\left(\hat{\boldsymbol{w}}^* + \sum_{i_0=1}^{i-1} \boldsymbol{l}_{i_0}\boldsymbol{u}_{i_0}\right) \geq \mathbb{E}_{\boldsymbol{\delta}} \mathsf{L}\left(\hat{\boldsymbol{w}}^* + \sum_{i_0=1}^{i} \boldsymbol{l}_{i_0}\boldsymbol{u}_{i_0}\right) + (c_i - 1)\boldsymbol{l}_i p_i / 2 - 2\xi > 0
$$

469 Where the last inequality holds as $\boldsymbol{l}_i > \frac{4\xi}{(c_i-1)p_i}$.

470 $\qquad\qquad\qquad\qquad\qquad\qquad\qquad\qquad\qquad\qquad\qquad\qquad\qquad\qquad\qquad\qquad\qquad\qquad\qquad\qquad\qquad$ $\square$

# C   Additional Figures for Section 4.2: Shift Exists Empirically

472 See Figure 17, Figure 18, and Figure 19.

Figure 17: Shift on asymmetric direction (DenseNet-100 on CIFAR-100), $\xi_{\boldsymbol{\delta}=1} = 0.119, \xi_{\boldsymbol{\delta}=0} = 0.439$

Figure 18: Shift on asymmetric direction (ResNet-164 on CIFAR-10), $\xi_{\boldsymbol{\delta}=0.5} = 0.0699$, $\xi_{\boldsymbol{\delta}=0} = 0.189$

Figure 19: Shift on symmetric direction (ResNet-110 on CIFAR-100), $\xi_{\boldsymbol{\delta}=1} = 0.0197$, $\xi_{\boldsymbol{\delta}=0} = 0.0431$

# D   Proof for Theorem 2

474 To prove Theorem 2, we will need the following concentration bound.

**Lemma 3** (Azuma's inequality). *Let $X_1, X_2, X_3, \ldots X_n$ be independent random variables satisfying $|X_i - \mathbb{E}[X_i]| \leq c_i$, for $1 \leq i \leq n$. We have the following bound for $X = \sum_{i=1}^{n} X_i$:*

$$
\Pr(|X - \mathbb{E}(X)| \geq \lambda) \leq 2e^{-\frac{\lambda^2}{2\sum_{i=1}^{n} c_i^2}}
$$

475 Let $p_{\min} \triangleq -\eta(a_- + a_+ + 2\nu)$, $p_{\max} \triangleq -\eta(b_- - \nu)$. Since $-a_- = ca_+$, we know $p_{\min} > $
476 $(c-1)\eta a_+ - 2\eta\nu$. First, we have the following bounds on the first step $w_0$.
477 **Lemma 4.** *For every $i \in [h]$, $w_0 \in [p_{\min}, p_{\max}]$.*

*Proof.* Since $w_0$ is the first step that SGD jumps from the flat side to the sharp side, denote the previous location as $w_{-1} < 0$. Since $w_{-1}$ is at the sharp side, we know that the gradient is $\nabla L(w_{-1}) \leq a_-$. Therefore, we have

$$w_0 = w_{-1} - \eta(\nabla L(w_{-1}) + \omega_{-1})$$

Where $\omega_{-1}$ is the noise bounded by $\nu$.

At the time when SGD jump from the flat side to sharp side, denote the target position as $w'_{-1}$. We know that $w'_{-1} \in [-\eta(a_+ + \nu), 0]$. Since the gradient on the sharp side is at most $a_-$, we know the next step is lower bounded by $-\eta(a_+ + 2\nu + a_-) = p_{\min} > 0$. In other words, SGD stays at the sharp side for only 1 iterations (this matches with our empirical observation, see e.g. Figure 5).

That means, the bound on $w'_{-1}$ can be applied to $w_{-1}$ as well, because they are the same iterate. By applying the upper and lower bound on $\nabla L(w_{-1})$, we get:

$$w_0 \geq -\eta(a_+ + \nu) - \eta(a_- + \nu) = p_{\min}$$

and also

$$w_0 \leq 0 - \eta(b_- - \nu) = p_{\max}$$

$\square$

Below we first define $T_{\min} \triangleq \left( \frac{-\sqrt{2}\nu \log^{1/2}(2\tau) + \sqrt{2\nu^2 \log(2\tau) - 4a_+(a_- + a_+ + 2\nu)}}{2a_+} \right)^2$, where $\tau$ is a constant with value to be set later. $T_{\min}$ satisfies the following inequality.

**Lemma 5.** $\forall t \leq T_{\min}, p_{\min} - t\eta a_+ - \sqrt{2t}\eta\nu \log^{1/2}(2\tau) \geq 0$.

*Proof.* By the definition of $p_{\min}$, we have

$$
\begin{aligned}
&-\eta(a_- + a_+ + 2\nu) - t\eta a_+ - \sqrt{2t}\eta\nu \log^{1/2}(2\tau) \geq 0 \\
\Leftarrow &(a_- + a_+ + 2\nu) + ta_+ + \sqrt{2t}\nu \log^{1/2}(2\tau) \leq 0 \\
\Leftarrow &(a_- + a_+ + 2\nu) + \Delta^2 a_+ + \sqrt{2}\Delta r \log^{1/2}(2\tau) \leq 0 \quad (\Delta \triangleq \sqrt{t}) \\
\Leftarrow &\Delta \in \left[ 0, \frac{-\sqrt{2}\nu \log^{1/2}(2\tau) + \sqrt{2\nu^2 \log(2\tau) - 4a_+(a_- + a_+ + 2\nu)}}{2a_+} \right] \\
\Leftarrow &t \leq \left( \frac{-\sqrt{2}\nu \log^{1/2}(2\tau) + \sqrt{2\nu^2 \log(2\tau) - 4a_+(a_- + a_+ + 2\nu)}}{2a_+} \right)^2
\end{aligned}
$$

$\square$

Now, we have the following theorem that says with decent probability, the minimum number of iterates on the flat side in $i$-th round is at least $T_{\min}$.

**Theorem 6.** *If we start at $w_0 \geq p_{\min}$, for every fixed $\tau > T_{\min}$, with probability at least $1 - \frac{T_{\min}}{\tau}$, we have $\forall t \leq T_{\min}, w_t > w_0 - t\eta a_+ - \sqrt{2t}\eta\nu \log^{1/2}(2\tau) \geq 0$.*

*Proof.* Define filtration $\mathcal{F}_t = \sigma\{\omega_0, \cdots, \omega_{t-1}\}$, where $\sigma\{\cdot\}$ denotes the sigma field. Define the event $\mathfrak{E}_T = \{\forall t \leq T, w_t > w_0 - t\eta a_+ - \sqrt{2t}\eta\nu \log^{1/2}(2\tau)\}$ and define $G_t = w_0 - w_t - t\eta a_+ + M$, where $M \triangleq (T_{\min} + 1)(w_0 + \nu + 2\eta a_+)$. Since we only consider the case $t \leq T_{\min}$, we have

$$G_t = w_0 - w_t - t\eta a_+ + (T_{\min} + 1)(w_0 + \nu + 2\eta a_+) > w_0 - w_t - t\eta a_+ + w_t + t\eta a_+ > 0$$

Therefore, $G_t$ is always positive. By SGD updating rule, we have

$$
\begin{aligned}
\mathbb{E}[G_{t+1}\mathbb{1}_{\mathfrak{E}_t} | \mathcal{F}_t] &= \mathbb{E}[(w_0 - w_{t+1} - (t+1)\eta a_+ + M)\mathbb{1}_{\mathfrak{E}_t} | \mathcal{F}_t] \\
&\leq \mathbb{E}[(w_0 - w_t + \eta\omega_t - t\eta a_+ + M)\mathbb{1}_{\mathfrak{E}_t} | \mathcal{F}_t] = w_0 - w_t - t\eta a_+ + M = G_t\mathbb{1}_{\mathfrak{E}_t}
\end{aligned} \tag{5}
$$

Since $\mathbb{1}_{\mathfrak{E}_t} \leq \mathbb{1}_{\mathfrak{E}_{t-1}}$, and $G_t$ is always positive, we have

$$G_t\mathbb{1}_{\mathfrak{E}_t} \leq G_t\mathbb{1}_{\mathfrak{E}_{t-1}} \tag{6}$$

503  Combining (5) and (6) together, we know $G_t \mathbb{1}_{\mathfrak{E}_{t-1}}$ is a supermartingale.

504  We can also bound the absolute value of the difference in every iteration:

$$
\begin{aligned}
&|G_{t+1}\mathbb{1}_{\mathfrak{E}_t} - \mathbb{E}[G_{t+1}\mathbb{1}_{\mathfrak{E}_t}|\mathcal{F}_t]| \\
=&|(w_0 - w_{t+1} - (t+1)\eta a_+ + M) - (w_0 - w_t - \nabla \mathsf{L}(w_t) - (t+1)\eta a_+ + M)|\mathcal{F}_t] \\
\leq & \eta\nu
\end{aligned}
$$

505  By Azuma's inequality, we get:

$$
\Pr\left(G_t\mathbb{1}_{\mathfrak{E}_{t-1}} - G_0 \geq \lambda\right) \leq 2e^{-\frac{\lambda^2}{2t\eta^2\nu^2}}
$$

506  That gives,

$$
\Pr\left(G_t\mathbb{1}_{\mathfrak{E}_{t-1}} - G_0 \geq \sqrt{2t}\eta\nu\log^{1/2}(2\tau)\right) \leq 1/\tau
$$

507  That means, if $\mathbb{1}_{\mathfrak{E}_{t-1}}$ holds, with probability at least $1 - 1/\tau$,

$$
w_0 - w_t - t\eta a_+ + M < \sqrt{2t}\eta\nu\log^{1/2}(2\tau) + G_0 = \sqrt{2t}\eta\nu\log^{1/2}(2\tau) + M
$$

508  Which gives

$$
w_t > w_0 - t\eta a_+ - \sqrt{2t}\eta\nu\log^{1/2}(2\tau)
$$

509  In other words, that means if $\mathbb{1}_{\mathfrak{E}_{t-1}}$ holds, then $\mathbb{1}_{\mathfrak{E}_t}$ also holds with probability at least $1 - 1/\tau$.

510  Therefore, if we are running $T_{\min}$ steps, we know that with probability at least $1 - \frac{T_{\min}}{\tau}$, $\mathbb{1}_{\mathfrak{E}_{T_{\min}}}$
511  holds. Therefore, by Lemma 5,

$$
\forall t \leq T_{\min}, w_t > w_0 - t\eta a_+ - \sqrt{2t}\eta\nu\log^{1/2}(2\tau) \geq p_{\min} - t\eta a_+ - \sqrt{2t}\eta\nu\log^{1/2}(2\tau) \geq 0 \quad \square
$$

512  Similarly, we define $T_{\max} \triangleq \left(\frac{-\sqrt{2}\nu\log^{1/2}(2\tau)+\sqrt{2\nu^2\log(2\tau)-4(b_- -\nu)b_+}}{2b_+}\right)^2$, which satisfies the fol-
513  lowing inequality.

514  **Lemma 7.** $p_{\max} - T_{\max}\eta b_+ - \sqrt{2T_{\max}}\eta\nu\log^{1/2}(2\tau) < 0$.

515  *Proof.* By the definition of $p_{\max}$, we want to show that

$$
(b_- - \nu) + T_{\max}b_+ + \sqrt{2T_{\max}}r\log^{1/2}(2\tau) \geq 0
$$

516  Which holds by the definition of $T_{\max}$. $\hspace{1cm}\square$

517  The Theorem below shows with decent probability, $T_{\max} - 1$ is an upper bound on the total number
518  of iterates on the flat side in the $i$-th round.

519  **Theorem 8.** *If $w_0 \leq p_{\max}$, with probability at least $1 - \frac{T_{\max}}{\tau}$, $w_{T_{\max}} < 0$.*

520  *Proof.* Define event $\mathfrak{E}'_T = \{\forall t \leq T, w_t < w_0 - t\eta b_+ + \sqrt{2t}\eta\nu\log^{1/2}(2\tau)\}$, and $G'_t = w_t + t\eta b_+ >$
521  $0$.

522  We have

$$
\begin{aligned}
&\mathbb{E}[G'_{t+1}\mathbb{1}_{\mathfrak{E}'_t}|\mathcal{F}_t] \\
=&\mathbb{E}[(w_{t+1} + (t+1)\eta b_+)\mathbb{1}_{\mathfrak{E}'_t}|\mathcal{F}_t] \\
\leq &\mathbb{E}[(w_t - \eta\omega_t + t\eta b_+)\mathbb{1}_{\mathfrak{E}'_t}|\mathcal{F}_t] \\
=&(w_t + t\eta b_+)\mathbb{1}_{\mathfrak{E}'_t} \\
=&G'_t\mathbb{1}_{\mathfrak{E}'_t}
\end{aligned}
$$

523  Moreover, we know $\mathbb{1}_{\mathfrak{E}'_t} \leq \mathbb{1}_{\mathfrak{E}'_{t-1}}$, which means $G'_t\mathbb{1}_{\mathfrak{E}'_t} \leq G'_t\mathbb{1}_{\mathfrak{E}'_{t-1}}$. So $G'_t\mathbb{1}_{\mathfrak{E}'_{t-1}}$ is a supermartingale.

524  We can also bound the absolute value of the difference in every iteration:

$$
\begin{aligned}
&|G'_{t+1}\mathbb{1}_{\mathfrak{E}'_t} - \mathbb{E}[G'_{t+1}\mathbb{1}_{\mathfrak{E}'_t}|\mathcal{F}_t]| \\
=&|(w_{t+1} + (t+1)\eta b_+) - (w_t - \eta\nabla\mathsf{L}(w_t) + (t+1)\eta b_+)|\mathcal{F}_t]| \\
\leq&\eta\nu
\end{aligned}
$$

525  Using Azuma inequality, we get

$$
\Pr\left(G'_t\mathbb{1}_{\mathfrak{E}'_{t-1}} - G'_0 \geq \sqrt{2t}\eta\nu\log^{1/2}(2\tau)\right) \leq 2e^{-\frac{t\eta^2\nu^2\log(2\tau)}{t\eta^2\nu^2}} = \frac{1}{\tau}
$$

526  That means, if $\mathbb{1}_{\mathfrak{E}'_{t-1}}$ holds, with probability at least $1 - 1/\tau$,

$$
w_t < w_0 - t\eta b_+ + \sqrt{2t}\eta\nu\log^{1/2}(2\tau)
$$

527  In other words, $\mathbb{1}_{\mathfrak{E}'_t}$ also holds. Therefore, if we are running $T_{\max}$ steps, we know that with probability
528  at least $1 - \frac{T_{\max}}{\tau}$, $\mathbb{1}_{\mathfrak{E}'_{T_{\max}}}$ holds. Therefore, by Lemma 7, we know

$$
w_{T_{\max}} < w_0 - T_{\max}\eta b_+ - \sqrt{2T_{\max}}\eta\nu\log^{1/2}(2\tau) < 0 \qquad\qquad \square
$$

529  **Remark.** To make sure Theorem 6 is not vacuous, we need to make sure that $T_{\min} \geq 1$. If we want
530  to make $T_{\min}$, say, at least 2, by Lemma 5, we have:

$$
p_{\min} - 2\eta a_+ - 2\eta\nu\log^{1/2}(2\tau) \geq 0
$$

531  Notice that $p_{\min} > (c-1)\eta a_+ - 2\eta\nu$, so we could solve the above inequality and get

$$
\begin{aligned}
&(c-1)\eta a_+ - 2\eta\nu - 2\eta a_+ - 2\eta\nu\log^{1/2}(2\tau) \geq 0 \\
\Rightarrow&\frac{(c-3)a_+ - 2\nu}{2\nu} \geq \log^{1/2}(2\tau) \\
\Rightarrow&\tau \leq \frac{e^{\left(\frac{(c-3)a_+}{2\nu}-1\right)^2}}{2}
\end{aligned}
$$

532  Since we assume that $c$ is a large constant and $a_+ \geq \nu$, so $\tau$ can be fairly large in order to make sure
533  $T_{\min} \geq 2$. We also know that $T_{\min} \leq \frac{-(a_- + a_+ + 2\nu)}{a_+} < c$.

534  On the other hand, by simple calculation, we know $T_{\max} \leq \frac{-(b_- - \nu)}{b_+} < c' < \frac{e^{c/3}}{6}$. Therefore, we
535  can always pick a $\tau$ such that $\frac{T_{\min} + T_{\max}}{\tau} \leq \frac{1}{2}$. So finally, we are ready to prove Theorem 2.

536  *Proof of Theorem 2.* By Lemma 4 and Theorem 8, $T_{\max}$ is an upper bound on the length of the $i$-th
537  round. By Theorem 6, we know that SGD will stay at flat side for at least $T_{\min}$ steps, and each step is
538  lower bounded by $w_t > w_0 - t\eta a_+ - \sqrt{2t}\eta\nu\log^{1/2}(2\tau)$, therefore we know that with probability
539  $1 - \frac{T_{\min} + T_{\max}}{\tau}$:

$$
\begin{aligned}
\frac{1}{T_i}\sum_{j=0}^{T_i} w_j^i &\geq \frac{1}{T_{\max}}\left(\sum_{t=0}^{T_{\min}}[w_0 - t\eta a_+ - \sqrt{2t}\eta\nu\log^{1/2}(2\tau)] - \eta(a_+ + \nu)\right) \\
&\geq \frac{1}{T_{\max}}\left(\eta a_+\frac{(T_{\min}+1)T_{\min}}{2} + \sqrt{2T_{\min}}\eta\nu\log^{1/2}(2\tau) - \eta(a_+ + \nu)\right) \\
&\geq \frac{T_{\min}^2}{T_{\max}}\eta a_+
\end{aligned}
$$

540  The above inequality discussed the scenario when Theorem 6 and Theorem 8 hold. If they do not hold,
541  which happens with probability at most $\frac{T_{\min} + T_{\max}}{\tau}$, we need to get lower bound for $\frac{1}{T_i}\sum_{j=0}^{T_i} w_j^i$.

Notice that by Lemma 4, we know that SGD stays at the sharp side for at most 1 iterate in each round, and also the iterates on the flat sides are always positive with $w_0 \geq p_{\min} > \eta(a_+ + \nu)$. Therefore, we have the following trivial bound:

$$\frac{1}{T_i} \sum_{j=0}^{T_i} w_j^i \geq \frac{-\eta(a_+ + \nu) + w_0}{2} > 0$$

Combining two cases together we get

$$\mathbb{E}\left[\frac{1}{T_i} \sum_{j=0}^{T_i} w_j^i\right] \geq \left(1 - \frac{T_{\min} + T_{\max}}{\tau}\right) \frac{T_{\min}^2}{T_{\max}} \eta a_+ + 0$$

Since we can pick $\tau$ s.t. $\frac{T_{\min} + T_{\max}}{\tau} \leq \frac{1}{2}$, we have

$$\mathbb{E}\left[\frac{1}{T_i} \sum_{j=0}^{T_i} w_j^i\right] \geq \frac{T_{\min}^2}{2T_{\max}} \eta a_+ \triangleq c_0 > 0 \qquad \square$$

# E Additional Figures in Section 6.1: No Bumps Between SGD and SWA Solutions

Asymmetric valley of ResNet-56 on CIFAR-10, $(\bar{r}, \underline{r}, p, c) = (3.7, 3.0, 0.016, 10)$. See Figure 20.

Asymmetric valley of ResNet-110 on CIFAR-10, $(\bar{r}, \underline{r}, p, c) = (5.3, 3.5, 0.0050, 11)$. See Figure 21.

Asymmetric valley of ResNet-164 on CIFAR-10, $(\bar{r}, \underline{r}, p, c) = (2.5, 2.0, 0.027, 4.3)$. See Figure 22.

Asymmetric valley of VGG-16 on CIFAR-10, $(\bar{r}, \underline{r}, p, c) = (5.6, 4.0, 0.0033, 30)$. See Figure 23.

Asymmetric valley of DenseNet-100 on CIFAR-10, $(\bar{r}, \underline{r}, p, c) = (13.0, 8.0, 0.0029, 7.4)$. See Figure 24

Asymmetric valley of ResNet-56 on CIFAR-100, $(\bar{r}, \underline{r}, p, c) = (11.0, 6.0, 0.034, 15)$. See Figure 25.

Asymmetric valley of ResNet-110 on CIFAR-100, $(\bar{r}, \underline{r}, p, c) = (7.5, 4.5, 0.053, 6.3)$. See Figure 26.

Asymmetric valley of ResNet-164 on CIFAR-100, $(\bar{r}, \underline{r}, p, c) = (11.0, 6.0, 0.012, 18)$. See Figure 27.

Asymmetric valley of VGG-16 on CIFAR-100, $(\bar{r}, \underline{r}, p, c) = (9.0, 6.0, 0.0084, 17)$. See Figure 28.

Asymmetric valley of ResNet-56 on SVHN, $(\bar{r}, \underline{r}, p, c) = (5.0, 4.0, 0.018, 15)$. See Figure 29.

Asymmetric valley of ResNet-110 on SVHN, $(\bar{r}, \underline{r}, p, c) = (4.5, 2.5, 0.010, 11)$. See Figure 30.

Asymmetric valley of ResNet-164 on SVHN, $(\bar{r}, \underline{r}, p, c) = (4.5, 2.5, 0.033, 7.0)$. See Figure 31.

Asymmetric valley of VGG-16 on SVHN, $(\bar{r}, \underline{r}, p, c) = (4.5, 2.5, 0.0043, 43)$. See Figure 32.

Asymmetric valley of ResNet-56 on STL-10, $(\bar{r}, \underline{r}, p, c) = (8.0, 5.0, 0.33, 2.4)$. See Figure 33.

Asymmetric valley of ResNet-110 on STL-10, $(\bar{r}, \underline{r}, p, c) = (11.0, 6.0, 0.51, 3.5)$. See Figure 34.

Asymmetric valley of ResNet-164 on STL-10, $(\bar{r}, \underline{r}, p, c) = (12.0, 7.0, 0.092, 16)$. See Figure 35.

Asymmetric valley of VGG-16 on STL-10, $(\bar{r}, \underline{r}, p, c) = (5.0, 3.0, 0.11, 12)$. See Figure 36.

Figure 20: SWA and SGD interpolation (ResNet-56 on CIFAR-10)

Figure 21: SWA and SGD interpolation (ResNet-110 on CIFAR-10)

Figure 22: SWA and SGD interpolation (ResNet-164 on CIFAR-10)

Figure 23: SWA and SGD interpolation (VGG-16 on CIFAR-10)

Figure 24: SWA and SGD interpolation (DenseNet-100 on CIFAR-10)

Figure 25: SWA and SGD interpolation (ResNet-56 on CIFAR-100)

Figure 26: SWA and SGD interpolation (ResNet-110 on CIFAR-100)

Figure 27: SWA and SGD interpolation (ResNet-164 on CIFAR-100)

Figure 28: SWA and SGD interpolation (VGG-16 on CIFAR-100)

Figure 29: SWA and SGD interpolation (ResNet-56 on SVHN)

Figure 30: SWA and SGD interpolation (ResNet-110 on SVHN)

Figure 31: SWA and SGD interpolation (ResNet-164 on SVHN)

Figure 32: SWA and SGD interpolation (VGG-16 on SVHN)

Figure 33: SWA and SGD interpolation (ResNet-56 on STL-10)

Figure 34: SWA and SGD interpolation (ResNet-110 on STL-10)

Figure 35: SWA and SGD interpolation (ResNet-164 on STL-10)

Figure 36: SWA and SGD interpolation (VGG-16 on STL-10)

 # F    Additional Figures in Section 6.1: SGD Averaging Generates Good Bias

Examples for asymmetric directions of ResNet-110 on CIFAR-100 in Figure 37.

Figure 37: The average of SGD has a bias on flat side (ResNet-110 on CIFAR-100).

 Examples for asymmetric directions of ResNet-164 on CIFAR-100 in Figure 38,

Figure 38: The average of SGD has a bias on flat side (ResNet-164 on CIFAR-100).

 Examples for asymmetric directions of ResNet-110 on CIFAR-10 in Figure 39.

Figure 39: The average of SGD has a bias on flat side (ResNet-110 on CIFAR-10).

## G    Batch size effect

Keskar et al. [32] observed that training with small batch size using SGD algorithm generalizes better than training with large batch size. They argue that it is because large batch SGD tends to converge to sharp minima, while small batch SGD generally converges to flat minima. Here we present a slightly different view that batch size has an influence on choosing sides of an asymmetric valley.

We use a PreResNet-164 trained on CIFAR-100 as an example. We first running SGD with a batch size of 128 for 200 epochs to find a solution (denoted as *Large batch solution*), and then contintue the training with batch size 32 for another 80 epoch to find a nearby solution (denoted as *Small batch solution*). The reason for fine-tune is that we hope the two solutions are not far from each other, and we want to show that small batch size ensures a bias towards flat side.

From the results shown in Figure 40, it is clear that the small batch solution has worse training accuracy but better test accuracy. Meanwhile, there is no 'bump' between these solutions which suggests they are in the same basin. Therefore, small batch SGD generalizes better because it could find a better biased solution in the asymmetric valley under our training scheme, not because it finds a different wider or flatter minimum.

Figure 40: Large and small minibatch interpolation(batch size 128 to 32 of PreResNet-164 on CIFAR-100)

## H    Batch Norm and Asymmetric Valleys

In this section, we present empirical evidences that the Batch Normalization (BN) [24] adopted by modern neural networks seems to be a major cause for asymmetric valleys.

**Directions on BN parameters are more asymmetric.**  For a given SGD solution, if we take a random direction where only the BN parameters have non-zero entries, and compare it with a random direction where only the non-BN parameters have non-zero entries, we observe that those BN-related directions are usually more asymmetric. The result with ResNet-110 on CIFAR-10 is shown in Figure 41, . As we can see, the Non-BN direction is sharp on both sides, but BN direction is flat on one side, and sharp on the other side. We also conducted trials with different networks and datasets, and obtained similar results (see Figure 42, 43 and 44).

Figure 41: BN and Non-BN directions through a local minimum of ResNet-110 on CIFAR-10.

Figure 42: BN and Non-BN directions through a local minimum of of ResNet-164 on CIFAR-10.

Figure 43: BN and Non-BN directions comparison of ResNet-110 on CIFAR-100

Figure 44: BN and Non-BN directions comparison of DenseNet-100 on CIFAR-100

Figure 45: SGD averaging on BN parameters give better test accuracy compared with SGD averaging on non-BN parameters.

Figure 46: Test accuracy of ResNet-8 with and without BN layers, after running weight averaging (SWA).

**SGD averaging is more effective on BN parameters.** By Theorem 1 and 2, we know that SGD averaging could lead to biased solutions on asymmetric directions with better generalization. If BN indeed creates many asymmetric directions, can we improve the model performance by only averaging the weights of BN layers?

Note that BN parameters only constitute a small fraction of the total model parameters, e.g., 1.41% in a ResNet-110. In the follow experiment on ResNet-110 for CIFAR-10, we perform SGD averaging only on BN parameters, denoted as SWA-BN; and also averaging randomly selected non-BN parameters of the same amount (1.41% of the total parameters), denoted as SWA-Non-BN. The results are shown in Figure 45. It can be observed that averaging only BN parameters (blue curve) is more effective than averaging non-BN parameters (green curve), although there is still a gap comparing to averaging all the weights (yellow curve).

Moreover, we also conduct experiments with two 8-layer ResNets on CIFAR-10, one with BN layers and one without. We choose shallow networks here as deeper models without BN can not be effectively trained.

As shown in figure 46, we start weight averaging at the 126-th epoch. Although in both networks, we observe an improvement in test accuracy after averaging, it is clear that the network with BN layers have larger improvement compared with the network without BN layers. This again indicates that SGD averaging is more effective on BN parameters.