[Reviews · NeurIPS 2019]

Reviewer 1



Summary: The authors analyse the energy landscape associated with the training of deep neural networks and introduce the concept of Asymmetric Valleys (AV), local minima that cannot be classified as sharp or flat local minima. AV are characterized by the presence of asymmetric directions along which the loss increases abruptly on one side and is almost flat on the other. The presence of AV in commonly used architectures is proven empirically by showing that asymmetric directions can be found with `decent probability'. The authors explain why SGD, with averaged updates, behaves well (in terms of the generalization properties of the trained model) in the proximity of AV. Strengths: The study of neural networks' energy landscape is a recent important topic. Existing analysis are often based on the flat-sharp classification of local minima and showing the presence of stationary points that escape such basic distinction is an important contribution. Moreover, the paper explains clearly and justify theoretically how the flatness and sharpness of minima affect the generalization properties of the trained model. The introduction contains a nice and accessible review of recent works on the topic. Weaknesses: It is hard to assess what is the contribution of the paper from a practical point of view as SGD automatically avoids possible training problems related to AV. On the theoretical side, the paper does not investigate deeply the relationship between AV and usual flat or sharp minima. For example, how are AV connected to each others and which properties of sharp/flat minima generalize to AV? Questions: - Under what conditions (on the network architecture) do AV appear? Is there an intuitive interpretation of why they can be found in the loss function associated with many `modern' neural networks? - Is the presence of AV restricted to the over-parameterized case? If yes, what happens in the under-parameterized situation? Which of the given theoretical properties extend to that case (as local minima cannot be expected to be equivalent in the under-parameterized case). - Do the structure of AV depend on the type of objective function used for training? What happens if a L-2 penalty term on the weights is added to the loss function? - Would it be possible to built a 2-dimensional analytical example of AV? - The averaged SGD performs well also in the case of convex loss. Is its good behaviour around a AV related to this? - In Section 3.2, it is claimed that AV can be found with `decent probability'. What is the order of magnitude of such probability? Does it depend on the complexity of the model? Does this mean that most of the minima are AV?

Reviewer 2



As you described in the paper, there are many studies on the generalization analysis of DNNs, including flatness/sharpness. In these studies, this paper focus on a novel concept of a loss landscape of DNNs. This work may be potentially useful, but the current version provides little explicit motivation for the proposed concept. What is the main contribution of analyzing asymmetric valleys compared with other concepts? What problems does this concept solve that cannot be solved with other theories? For example, flatness typically seems to include asymmetric valleys.

Reviewer 3



The paper provides interesting insights about the existence of asymmetric valleys in deep neural networks and claims that asymmetric valleys can lead to better generalization. The authors impose slightly non-standard strong assumptions, but empirically demonstrate that these assumptions are in fact practical and not difficult to achieve in practice. The paper is extremely well written and easy to read. However, there are few issues which concern me: 1) The authors state that the biased solution at an asymmetric valley is better than the biased solution at an asymmetric valley. How does it compare to a unbiased solution at a symmetric valley? It is not clear how often do these algorithms end up in asymmetric valleys and if using a biased solution is a good idea when we land up at symmetric valleys 2) Do the authors run optimize for learning rates for SGD? Additionally, what decay schedules have the authors experimented with? 3) Why do authors run SGD from the SWA solutions? While this provides evidence that if are at an asymmetric valley then SWA performs better than SGD, however how often does SGD end up in the same neighborhood (valley) as SWA solution and the relative generalization guarantees are unclear? Post-rebuttal: Having read the author's rebuttal, I would like to change my review from weak accept to an accept

[Author Response · NeurIPS 2019]

First of all, we would like to thank all reviewers for the insightful comments and suggestions! Reviewers have also raised many inspiring questions on asymmetric valleys (AVs), most of which we have addressed in this rebuttal. But for some of them (like what network structure or loss function tend to cause AVs, and what other new theoretical results could be obtained based on AVs), we may NOT have satisfying answers yet. However, this is perhaps one of the most valuable contributions of the paper – spawning new research problems and inspiring future research.

## General Response

**Significance and Novelty.** Optimization landscape analysis is an important research topic in deep learning. To the best of our knowledge, this work for the first time introduces and formally defines AV. This goes beyond simply characterizing a local minimum by sharp or flat, which are popular terminology in the literature. The concept of AV leads to new results which may NOT be possible to derive based on existing terminology (see our next response).

**What can be explained by AVs but not symmetric valleys (SVs).** Here we give two examples (more details can be found in Sec 6.1 and 6.2): (1) Recent work [25,5,51] found that stochastic weight averaging (SWA) over iterations leads to HIGHER TRAINING LOSS but lower test error. If local minimum are SVs, then by simple concentration arguments, SWA should lead to LOWER TRAINING LOSS! In contrast, AVs gives a nice intuitive explanation for those interesting observations, and we have provided rigid theoretical analysis. (2) Recent work [12,43] observed that the local minimum of deep networks are well connected, meaning that a wide minimum and a sharp minimum could be in fact from the SAME basin. This seemingly contradictory observation can be well explained by AVs, but not SVs.

**Are AVs prevalent?** Yes. In our experiments, we can always find asymmetric directions at every local minimum that SGD finds, for all networks and datasets. To be conservative, we used the word "decent probability" in our paper.

Figure 1: An AV in 2D-MLP

**Do AVs only appear in deep nets? What about 2-D loss surfaces?** Apart from the SOTA networks stated in our paper, we also conduct experiments on a simple MLP in Appendix. Following Reviewer 2's suggestion, we also tried a 2D-MLP (1 single neuron with its weight, bias and sigmoid activation) experiment: data is drown from two 1D Gaussian distributions, forming a binary classification problem. It turns out that in such a simple case we can also find AVs (shown in Figure 1).

## To Reviewer #1

Thanks a lot for your inspiring questions. As AV is a novel concept, we are not able to study *all* its specific properties in this work, and we do to have answers to several of your questions yet. But we believe that our work provides a new perspective to understand the loss landscape of deep networks, and may inspire many interesting future research topics.

**SGD automatically avoids the training problem?** In fact, SGD does not automatically lead to desired solutions for AVs, but averaged SGD does (Theorem 2).

**AV and objective function.** We believe that the structure of AV depends on the objective function. The whole story is quite complicated. But adding a L-2 penalty (as you suggested) seems to have little effect on AVs: we could also find AVs, and averaged SGD still gives better performance. Studying the relation between AV and objective function (and network architectures) is an interesting future research direction.

**What leads to AVs?** The reason of why AV exist is not fully understand yet, but we believe batch normalization is one of the important reasons (please refer to Appendix H). We leave it as the future work.

## To Reviewer #2

**Motivation and contribution?** Our result is important because the notion of AVs can be used for explaining many interesting observations (e.g., [25,5,51]) which can not be well explained by existing concepts. Please refer to the "General Response" above.

**Flatness includes AVs?** This is true if we still simply characterize both asymmetric and symmetric valleys by flatness, without differentiating SVs and AVs. However, without introducing AVs, we will not be able to obtain the theoretical results on bias and generalization, and those observations made by [25,5,51] cannot be well explained.

## To Reviewer #3

**Bias and SVs?** In fact, bias is good for AVs, but not for SVs. Fortunately, SGD averaging automatically generate bias for AVs (Theorem 2) and less bias for SVs (following a simple concentration argument). As our paper is on AVs, we dismissed the discussion on SVs. **Learning rate.** We follow all the hyperparameter configurations used in [25].

**Are SGD and SWA end up in the same neighborhood?** As SWA is the averaged solution of SGD iterates, they are located in the same neighborhood under mild conditions. Therefore, our theoretical generalization guarantee can be ensured. In our experiment, we further run SGD from SWA because we want to find a solution that clearly has lower training loss than SWA, but has a higher test loss, thus validating our theoretical results. Also notice that empirically SWA still generalizes better than SGD even when they are not in the same local basin (see e.g. [25]).

[Meta-Review · NeurIPS 2019]

This paper introduces the concept of asymmetric valleys (AVs), studies the generalization performance of the flat side, and provides an algorithm that is biased toward the flat side. Initially the paper received mixed reviews, with two positive and one negative reviews, the latter one arguing the need for more explicit motivation for AVs. The rebuttal successfully addressed a large number of questions raised by the reviewers, and the negative reviewer updated his/her score. Upon discussion, the reviewers agreed that the paper should be accepted. Overall, this paper addresses a very important and timely problem (understanding the relationships between landscape, optimization and generalization).